# Erythrocytes as Carriers of Therapeutic Enzymes

**DOI:** 10.3390/pharmaceutics12050435

**Published:** 2020-05-08

**Authors:** Bridget E. Bax

**Affiliations:** Molecular and Clinical Sciences, St. George’s, University of London, London SW17 0RE, UK; bebax@sgul.ac.uk; Tel.: +44-(0)208-266-6836

**Keywords:** carrier erythrocytes, erythrocyte carriers, enzyme replacement therapies, therapeutic enzymes, thrombolytic therapy, drug delivery, erythrocyte bioreactor

## Abstract

Therapeutic enzymes are administered for the treatment of a wide variety of diseases. They exert their effects through binding with a high affinity and specificity to disease-causing substrates to catalyze their conversion to a non-noxious product, to induce an advantageous physiological change. However, the metabolic and clinical efficacies of parenterally or intramuscularly administered therapeutic enzymes are very often limited by short circulatory half-lives and hypersensitive and immunogenic reactions. Over the past five decades, the erythrocyte carrier has been extensively studied as a strategy for overcoming these limitations and increasing therapeutic efficacy. This review examines the rationale for the different therapeutic strategies that have been applied to erythrocyte-mediated enzyme therapy. These strategies include their application as circulating bioreactors, targeting the monocyte–macrophage system, the coupling of enzymes to the surface of the erythrocyte and the engineering of CD34^+^ hematopoietic precursor cells for the expression of therapeutic enzymes. An overview of the diverse biomedical applications for which they have been investigated is also provided, including the detoxification of exogenous chemicals, thrombolytic therapy, enzyme replacement therapy for metabolic diseases and antitumor therapy.

## 1. Introduction

Therapeutic enzymes are biocatalyst drugs that bind to target substrates with a high affinity and specificity, catalyzing their conversion into their relevant products. The past five decades have seen the development of therapeutic enzymes for treating a wide range of medical conditions, including inherited enzyme deficiency disorders, acute poisoning, digestive disorders, cancer and cardiovascular diseases. Chemical modifications of the native enzyme (e.g., conjugation with polyethylene glycol) are often employed in the manufacturing process to increase protein stability, decrease immunogenicity, reduce renal ultrafiltration and in some cases, to enable targeting of the enzyme to the appropriate cellular compartment [1]. However, despite these strategies, the metabolic and clinical efficacy of parenterally or intramuscularly administered therapeutic enzymes is still limited. This is principally due to short circulatory half-lives, hypersensitivity reactions and immunogenicity. Immune responses are influenced by several factors including the biophysical characteristics of the protein, the route of delivery, the degree of exposure and the use of immunosuppressive agents during administration. Host factors may also play a part, for example patients with congenital enzyme deficiencies may fail to recognize a therapeutic protein as “self” and may be more likely to mount an immune response during the administration of therapeutic replacement enzyme [2]. The consequences of an immune reaction to a therapeutic enzyme range from a transient appearance of antibodies, without any clinical sequel, to severe life-threatening conditions.

The erythrocyte carrier has been extensively studied as a strategy for overcoming these limitations and increasing therapeutic efficacy. For a majority of the therapeutic applications investigated, the ability of the cell to reseal after creating pores in the membrane has been exploited for the purpose of introducing therapeutic agents [3,4,5]. The resealed erythrocyte is biocompatible and, in the human, has a normal *in vivo* circulating half-life of 19–29 days, and thus raises the potential to extend the half-life of encapsulated enzymes through the avoidance of plasma clearance due to the action of proteases, anti-enzyme antibodies and renal clearance, and through minimizing immune reactions.

The aim of this article is to provide a review of the available literature relating to *in vitro*, preclinical and clinical studies of the erythrocyte as a vehicle for therapeutic enzymes.

## 2. Therapeutic Strategies

Four main therapeutic strategies have been applied to the erythrocyte carrier in terms of a vehicle for therapeutic enzymes. In the first, enzyme-loaded erythrocytes can be employed as circulating bioreactors whereby blood-based molecules diffuse into the erythrocyte and are degraded. This strategy is relevant to the treatment of congenital enzyme-deficiencies where a pathologically elevated plasma metabolite is able to permeate erythrocyte membrane and undergo metabolism to the normal product [6]. Also relevant to this therapeutic strategy is the depletion of plasma metabolites for the treatment of auxotrophic tumors and detoxification after exposure to toxic chemicals, Figure 1.

The second strategy involves the targeting of encapsulated enzymes to the monocyte–macrophage system of the spleen, liver and bone marrow, through exploiting the natural sites of erythrophagocytosis [7]. The loss of phospholipid asymmetry and the exposure of phosphatidylserine on the surface of the plasma membrane of senescence erythrocytes induces erythrocyte sequestration by macrophages. As erythrocyte degradation occurs in the lysosomal compartment of macrophages, it is proposed that erythrocyte encapsulated enzymes can potentially target macromolecules that accumulate in the lysosomal compartment as a result of congenital enzyme-deficiencies, Figure 2.

The third strategy involves the coupling of therapeutic enzymes to the erythrocyte surface with the aim of improving the therapeutic profile, Figure 3.

The fourth strategy is a relatively new approach. This is based on the engineering of CD34^+^ hematopoietic precursor cells to express therapeutic enzymes and then their subsequent differentiation until the nucleus is ejected, resulting in the mature reticulocyte, Figure 4. This strategy has potential applications for both the metabolism of plasma metabolites and targeting of the monocyte–macrophage system.

## 3. Therapeutic Applications

Over the past five decades, the erythrocyte carrier has been extensively investigated as a potential modality for targeting a wide range of therapeutic applications. In relation to carriers of enzymes, these applications can broadly be divided into four key areas of research interest: exogeneous chemical detoxification, thrombolytic therapy, treatments for metabolic diseases and antitumor therapies. Although a majority of these studies have not translated beyond the pre-clinical stage, a few are currently undergoing clinical development. The following provides a discussion of the applications investigated under these four research areas.

### 3.1. Detoxification of Exogenous Chemicals

The utility of the erythrocyte as a carrier of specific enzymatic antidotes against chemical intoxicants has been investigated by a number of groups. Investigations into this therapeutic application have focused on encapsulating the relevant enzyme within the erythrocyte. Way et al. first established the potential of the erythrocyte carrier as antidote candidate for cyanide intoxication [8,9]. Cyanide is a rapidly lethal poison and exerts its toxic effect by blocking the mitochondrial respiration chain and the formation of intracellular adenosine triphosphate through binding to cytochrome-c oxidase, the terminal enzyme complex of the respiratory chain in complex IV. Sodium thiosulfate is one of the antidotes used to combat intoxication and works by acting as a sulfur donor for the mitochondrial enzyme, thiosulfate sulfotransferase (rhodanase, EC 2.8.1.1) [10]. Sodium thiosulfate does not readily permeate cell membranes and therefore is not able to distribute to sites of thiosulfate sulfotransferase or cyanide localization; this provided the rationale for investigating the erythrocyte carrier as an alternative approach to cyanide antagonism. *In vivo* studies in mice demonstrated that erythrocytes containing sodium thiosulfate and rhodanase could rapidly metabolize cyanide to the less toxic thiocyanate and antagonize the effects of a lethal dose of potassium cyanide [11,12,13]. Moreover, by replacing sodium thiosulfate with butanethiosulfonate, a more reactive sulfur donor substrate, an enhanced protective effect against cyanide was found [14].

The application of the erythrocyte carrier as an antagonist of the lethal effects of parathion, a once widely used agricultural organophosphorous insecticide was also investigated as an alternative antidote approach to paraoxon intoxication. The toxicity of parathion is attributed to its *in vivo* metabolism to paraoxon which inhibits acetylcholinesterase, leading to an accumulation of acetylcholine and ultimately altering cholinergic synaptic transmission at neuroeffector junctions, at skeletal myoneural junctions and autonomic ganglia in the central nervous system [15]. Two antidotes for parathion poisoning are atropine, a competitive antagonist of acetylcholine at the muscarinic receptor and pralidoxime, which regenerates acetylcholinesterase [16]. However, neither of these antidotes are able to degrade parathion. *In vivo* studies in mice investigated the efficacy of erythrocyte encapsulated phosphotriesterase (EC 3.1.8.1) in antagonizing the lethal effects of paraoxon through its hydrolysis to the less toxic 4-nitrophenol and diethylphosphate [17]. The results indicated that although the phosphotriesterase-loaded erythrocytes were more effective than the classic antidotal combination of atropine and pralidoxime, a combination of the loaded erythrocytes with the classic antidot, provided a 1000-fold protection against paraoxon. The same group also investigated the application of erythrocyte encapsulated recombinant paraoxonase as an approach to directly hydrolyze paraoxon; treated mice showed no signs of intoxication at paraoxon dose levels that were lethal when using the classical antidotal combination of atropine and pralidoxime. Moreover, erythrocyte encapsulated paraoxonase, in combination with the classic antidotal combination, provided the highest antidotal efficacy ever reported against any chemical toxicant [18].

Another category of detoxifying enzymes that have been investigated are those associated with the metabolism of ethanol and methanol. Ethanol detoxification requires two enzymic reactions: the oxidation of ethanol to acetaldehyde by alcohol dehydrogenase (EC. 1.1.1.1), followed by the oxidation acetaldehyde to acetate by aldehyde dehydrogenase (EC 1.2.1.5). Chronic alcohol consumption decreases acetaldehyde oxidation, either due to decreased aldehyde dehydrogenase activity or impaired mitochondrial function. The application of the erythrocyte carrier as an alcohol detoxifier was first proposed by Magnani *et al*. They demonstrated that mice administered acetaldehyde dehydrogenase-loaded erythrocytes intraperitoneally had 35% less blood acetaldehyde compared to controls, one hour after receiving an acute dose of ethanol [19]. Lizano et al. investigated the co-encapsulation of alcohol dehydrogenase and aldehyde dehydrogenase in human erythrocytes, demonstrating their superior ability to metabolize ethanol *in vitro*, compared to erythrocytes loaded with alcohol dehydrogenase alone [20]. *In vivo* studies in mice receiving acute doses of ethanol and treated with co-encapsulated enzymes showed blood ethanol to be eliminated at rates of 1.7 mmol/L to 4 mmol/L loaded erythrocytes/hour [21,22]. However, these rates of plasma ethanol clearance were lower by an order of magnitude than those expected from the activities of encapsulated enzymes. By employing a mathematical modeling approach and then conducting a subsequent *in vitro* study, Alexandrovich et al. were able to theorize and then demonstrate the rate limiting step of external ethanol oxidation. They found this was due to the rate of nicotinamide–adenine dinucleotide (NAD+) generation in erythrocyte glycolysis, rather than the activities of the loaded enzymes. By supplementing the erythrocytes with NAD+ and pyruvate they were able to demonstrate an elimination of 17 mmol ethanol/L loaded erythrocytes/hour [23].

In mammalian species, methanol is metabolized to formaldehyde via alcohol dehydrogenase, followed by the conversion of formaldehyde into formic acid via aldehyde dehydrogenase. Formic acid metabolism is mediated through a tetrahydrofolate-dependent pathway by folate-dependent enzymes. Humans have 60% less liver folate concentrations compared to mice and rats, and for this reason humans are more sensitive to methanol poisoning [24]. Specifically, formic acid inhibits mitochondrial cytochrome c oxidase, leading to cellular hypoxia and metabolic acidosis. Magnani et al. investigated the application of erythrocyte-encapsulated methylotrophic yeast alcohol oxidase (EC 1.1.3.13) as an approach to the detoxification of methanol. *In vivo* studies showed that two hours following an acute dose of methanol (0.7 g/kg), mice that had received enzyme-loaded erythrocytes had 50% less blood methanol compared to controls, with antagonism persisting for at least one week [25]. On the basis that formate dehydrogenase (EC 1.2.1.2) converts formate into CO_2_ in the presence of NAD, Muthuvel et al. administered formate dehydrogenase-loaded erythrocytes to methanol-intoxicated folate-deficient rats, which were pre-treated with carbicarb to correct the metabolic acidosis, and demonstrated a marked elimination of formate [26].

A detoxifying enzyme that has been investigated in relation to lead poisoning is δ-aminolevulinic acid dehydratase (E.C. 4.2.1.24). Lead is a potent inhibitor of δ-aminolevulinic acid dehydratase through its displacement of zinc from the enzyme’s active site. The resulting inactivation leads to an accumulation of its substrate, aminolevulinic acid, and this has been shown to have a neuropathogenic effect [27]. The *in vitro* and *in vivo* mouse studies of Bustos et al. showed that it was possible to correct the defective δ-aminolevulinic acid dehydratase activity in erythrocytes by encapsulating exogenous human enzyme, thus providing a rationale for this approach for treating lead intoxication [28,29]. In the first clinical application, 100 mL autologous erythrocytes loaded with 25,600 units of human δ-aminolevulinic acid dehydratase were administered to a single patient with a history of chronic lead intoxication, producing a reduction in blood lead concentration and decrease in urinary porphyrin excretion, which was sustained for a period of four years [30].

Biochemical decompression has been proposed as a method for reducing the amount of time required for deep-sea divers to return to the surface. Divers breathing H_2_/O_2_ mixtures would be administered hydrogenase (EC 1.12.1.4), thus accelerating decompression through the removal of excess hydrogen from the tissues. The application of erythrocyte encapsulated hydrogenase as an approach to converting dissolved hydrogen gas into non-gaseous forms was investigated by Axley et al. Hydrogenase activities of 7-µmol H_2_/minute/mL packed human erythrocytes were successfully encapsulated, however the cells were not able to consume hydrogen due to the enzyme’s dependence on NAD for activity. The co-encapsulation of FAD with hydrogenase was shown to overcome this [31].

### 3.2. Thrombolytic Therapy

The application of erythrocytes as a strategy for thrombolytic therapy has been explored by several groups. Thrombolytics (or fibrinolytics), also known as plasminogen activators, facilitate the removal of pre-existing thrombotic clots through degrading the fibrin meshwork into plasmin. However, their application is limited by their rapid clearance and inactivation by plasminogen activator inhibitor-1, the risk of bleeding caused by dissolution of hemostatic clots and filtration into extravascular tissues, such as the central nervous system, causing toxicity [32]. The rationale for the erythrocyte encapsulation of fibrinolytics was based on addressing these issues with the study of Delahousse et al. who successfully demonstrated the encapsulation of the fibrinolytic enzymes urokinase (EC 3.4.21.31), streptokinase (EC 3.4.99.0) and recombinant tissue plasminogen activator (tPA; EC 3.4.21.68) in human erythrocytes. However, *in vitro* stability studies using human erythrocytes and *in vivo* studies in the mouse were not able to support the objective of a sustained half-life and demonstrated a rapid destruction of the encapsulated enzymes [33]. Flynn and coworkers reported that *Aspergillus oryzae* brinase (EC. 3.4.99) encapsulated into rabbit erythrocytes was able to lyse clotted blood *in vitro* [34]. The coupling of fibrinolytics to the surface of the erythrocyte carrier was proposed as a preferable alternative to encapsulation [35]. Murciano et al. investigated the conjugation of tPA to erythrocytes as a strategy for thromboprophylaxis, by employing mouse and rat models of venous and arterial thrombosis. Following intravenous injection, the fibrinolytic activity of tPA conjugated to erythrocytes was shown to persist in the circulation at least tenfold longer than that of free tPA. Free tPA was able to lyse pulmonary clots that had lodged before administration, but not those that lodged after injection. Erythrocyte conjugated-tPA, however was more selective in lysing nascent over preexisting pulmonary emboli and arterial clots, an effect that is most likely a result of restricting the diffusion of tPA into fibrin [36]. The utility of this approach in preventing cerebrovascular thrombosis was also examined using mouse and rat models of cerebrovascular thromboembolism and ischemia, with the results indicating that erythrocyte-conjugated tPA was an effective thromboprophylaxis, whereas free tPA was ineffective up to a 10-fold higher dose [37].

Worthy of mention is the more recent studies investigating the erythrocyte carrier as a theranostic nanoplatform system. This system consists of vesicles derived from erythrocytes that encapsulate the NIR fluorophore, indocyanine green and of relevance to this review, the conjugation of tPA to the vesicle surface. These constructs are referred to as NIR erythrocyte-derived transducers (NETS). *In vitro* studies employing a clot model demonstrated the dual functionality of NETS in NIR imaging and clot lysis [38].

### 3.3. Enzyme Replacement for Metabolic Diseases

The application of the erythrocyte as a carrier of enzymes for the treatment of inherited metabolic diseases and other metabolic disturbances has generated a large amount of interest from those working in a field where there are enormous unmet needs. The following provides a discussion of the metabolic disorders to which the erythrocyte carrier has been applied. With the exception of phenylketonuria, these therapeutic applications have focused on the use of encapsulated enzyme.

#### 3.3.1. Lysosomal Storage Disorders

Ihler et al. first proposed the erythrocyte carrier as a strategy for targeting enzyme replacement for the treatment of disorders of sphingolipid catabolism, and successfully demonstrated the erythrocyte encapsulation of β-glucosidase (EC.3.2.1.45) and β-galactosidase (EC 3.2.1.23) [3]. Sphingolipidoses are a subgroup of lysosomal storage disorders characterized by an abnormal accumulation of phospholipids that have a sphingosine group. The rationale for the therapeutic strategy is based on the erythrocyte being naturally sequestered by cells of the monocyte–macrophage system and subsequent lysosomal degradation at the subcellular site where the enzyme defect manifests, thereby facilitating the delivery of the encapsulated enzyme to the site of phospholipid accumulation. Deloach and coworkers demonstrated that human erythrocytes-loaded with *E. coli* β-galactosidase could be phagocytosed by bone marrow macrophages that had been matured *in vitro*. Electron microscopy revealed that intact or partly degraded erythrocytes were localized in intracellular vacuoles, which were presumed to be phagolysosomes. The activity of the untaken β-galactosidase disappeared with a half-life of 15–30 h, which was consistent with the enzyme being degraded within the lysosomes [39]. Studies comparing the fate of erythrocyte-encapsulated bovine β-glucuronidase (EC 3.2.1.31) and free enzyme administered intravenously to β-glucuronidase-deficient mice demonstrated that erythrocyte encapsulated enzyme was retained four times longer in the circulation, five-fold longer in hepatic tissues and was more efficiently delivered to a number of other tissues [40]. In 1977, Beulter et al. reported the first clinical application of the erythrocyte carrier for the treatment of Gaucher disease (Online Mendelian inheritance in Man (OMIM) # 230800, 230900, 231000). Gaucher disease is caused by mutations in the *GBA* gene which encodes for the enzyme β-glucocerebrosidase (also referred to as β-glucosidase), which catalyzes the hydrolysis of glucosylceramide into ceramide and glucose. A deficiency in the enzyme leads to an accumulation of glucosylceramide in the lysosomes of macrophages in the liver, spleen and bone marrow. The disease is classified into three clinical forms on the basis of neurological involvement: type I (non-neuronopathic), type II (acute neuronopathic) and type III (subacute neuronopathic) with all three having symptoms of hepatosplenomegaly, anemia and orthopedic complications [41]. In the first clinical application, partially purified glucocerebrosidase isolated from the human placenta was encapsulated into erythrocytes and administered to a single patient with advanced type I Gaucher disease. Although there were no conclusive findings that enzyme infusion had been beneficial, this study was able to demonstrate the safety of repeated administrations of enzyme encapsulation in erythrocytes [42]. Studies investigating the *in vitro* uptake of glucocerebrosidase by Gaucher patient monocytes demonstrated that enzyme loaded erythrocytes coated with human IgG and agglutinated with anti-human serum were more avidly phagocytosed compared to uncoated loaded erythrocytes. In addition, the uptake of enzyme via IgG-coated erythrocytes, was sufficient to normalize cellular glucocerebrosidase activities for at least 18 h [43]. Bax et al. investigated the encapsulation of mannose-terminated glucocerebrosidase (Alglucerase), a licensed pharmaceutical enzyme preparation available for the treatment of Gaucher’s disease. This enzyme preparation has a plasma half-life of 3.6–10.4 min and thus, regular infusions are required to maintain therapeutic levels. Alglucerase encapsulation was found to be maximized by employing the more concentrated pharmaceutical preparation and doubling the hypo-osmotic dialysis time to 180 min [44,45]. Despite the erythrocyte carrier offering a potential solution to the rapid plasma clearance and the prospect of reducing therapy costs and the number of therapeutic interventions, this approach has not been advanced in the clinical setting. The availability of a number of approved and effective enzyme replacement therapies for Gaucher disease that can be administered in the patient’s home no doubt would have contributed to the lack of incentive in developing the erythrocyte carrier further for this disease.

#### 3.3.2. Hyperammonemia

A number of groups have investigated the use of the erythrocyte carrier as an alternative approach to treating metabolic disturbances characterized by an excess of ammonia in the blood. Ammonia is produced by intestinal bacterial flora or as a metabolic byproduct of amino acid catabolism acid and other compounds which contain nitrogen. Normally, ammonia is efficiently handled by the conversion to urea via the urea cycle in the liver or by the reaction catalyzed by glutamine synthetase (EC 6.3.1.2). Congenital deficiencies of enzymes in the urea cycle or diseases that lead to acute or chronic liver failure can result in hyperammonemia, leading to nervous system disturbances, hepatic encephalopathy, coma and death. The metabolic capability of the erythrocyte carrier in reducing blood ammonia concentrations has been investigated by the encapsulation of two enzymes. In the first approach, l-glutamate dehydrogenase (EC. 1.4.1.3) which catalyzes the formation of l-glutamic acid from α-ketoglutarate and ammonium in the presence of NADPH was investigated. The second approach utilized glutamine synthetase (EC. 6.3.1.2) which catalyzes the formation of l-glutamine from l-glutamic acid and ammonium in the presence of ATP. Sanz et al. explored the former approach by examining the capability of encapsulated glutamate dehydrogenase in removing extracellular ammonia *in vitro* by incubating human loaded erythrocytes in buffer containing ammonium chloride, α-ketoglutarate and NADPH as substrates. After 24 h of incubation, the ammonia concentration was 15 µmol/L, this representing 15% of the zero time concentration and therefore providing support for this approach [46]. Subsequent *in vivo* studies were performed in mice where hyperammonemia was induced through the intraperitoneal injection of 33 U of urease/Kg of body weight. One hour after urease administration the blood concentration of ammonia was 1333 µmol/L, and this declined to 135 µmol/L by 25 hours. By administering erythrocyte-encapsulated glutamate dehydrogenase to these mice, the blood concentration of ammonia could be reduced by 50%, two hours after urease injection. The ammonia concentration remained unchanged in control mice that received native washed erythrocytes [47]. Using the second approach, Venediktova et al. examined the capability of erythrocytes loaded with glutamine synthetase to eliminate ammonium from the blood of hyperammonemic mice. Blood ammonium concentrations were shown to decrease by approximately 2-fold, 30 min after injection of the loaded cells and 4-fold after 120 min, compared to ammonium concentration in control mice [48,49]. However, after this time, the blood ammonium concentration decreased approximately at the same rate in both the experimental and the control animals. To provide insight into this limitation, Protasov and coworkers developed mathematical models of the erythrocyte carriers by focusing on the glycolytic pathway reactions and enzymes utilizing ammonium. They concluded that the efficiency and duration of the erythrocyte carrier function was limited by the low membrane permeability for α-ketoglutarate and glutamate. To compensate the effect of this, the authors proposed the co-encapsulation of glutamate dehydrogenase and alanine aminotransferase (EC 2.6.1.2). This approach created a metabolic pathway where glutamic acid produced by the glutamate dehydrogenase reaction was converted into α-ketoglutarate by alanine aminotransferase, with α-ketoglutarate then entering the glutamate dehydrogenase reaction. Therefore, α-ketoglutarate and L-glutamic acid are produced and consumed in a cyclical process and hence the system will independent of their transport. The conclusions of the theoretical study were verified experimentally, demonstrating the removal of ammonium *in vitro* at the rate of 1.5 mmol/h/ L human loaded erythrocytes and in a *in vivo* model of hyperammonemia in mice at the rate of 2.0 mmol/h/L loaded erythrocytes [50].

An example of a urea cycle disorder that causes hyperammonemia is arginase-1 deficiency (OMIM # 207800). This autosomal recessive disorder is caused by a mutation in the *ARG1* gene. Arginase (EC 3.5.3.1) normally controls the final step in the urea cycle through the hydrolysis of arginine to urea and ornithine. Although a deficiency leads to an accumulation of arginine and ammonia in the blood and cerebrospinal fluid, it is the accumulation of arginine which predominantly contributes to the pathological abnormality, with affected individuals experiencing development delay, growth retardation and epileptic seizures [51]. Current treatments focus on reducing plasma ammonia and arginine concentrations and reducing dietary nitrogen. Adriaenssens et al. explored the capacity of arginase-loaded erythrocytes from patients with arginase deficiency to metabolize extracellular arginine *in vitro*. They demonstrated that when the enzyme-loaded erythrocytes were incubated in plasma containing 45 to 375 µmol/L arginine, the extracellular concentration of arginine was reduced by 34% to 53% after one hour [52].

The utility of the erythrocyte as a carrier of urease (EC 3.5.1.5) for reducing blood concentrations of urea in patients with chronic renal failure has been a long-term interest of Baysal et al. [53]. The rational for this approach is based on urease catabolizing urea into ammonia and bicarbonate, followed by the catalytic conversion of ammonia and erythrocyte pyruvate into alanine by alanine dehydrogenase (EC 1.4.1.1). The ultimate aim is to negate the requirement for hemodialysis and continuous ambulatory peritoneal dialysis. Urease and alanine dehydrogenase and their pegylated counterparts PEG–urease/PEG–alanine dehydrogenase were successfully encapsulated in human erythrocytes [54,55]. *In vivo* studies in sheep employing erythrocytes containing PEG–urease and PEG–alanine dehydrogenase at an activity unit ratio of 3:9 demonstrated a urea reduction of 51.6 mg/L/24 h. Compared to unencapsulated PEG–urease/PEG–alanine dehydrogenase, which was able to reduce blood urea concentration by 21.7–61.6 mg/L for two days, the same dose encapsulated in erythrocytes sustained these reductions for six days, thus demonstrating the utility of this system in sustaining enzyme activity [56].

#### 3.3.3. Hyperglycemia

The exploitation of enzyme-loaded erythrocytes in the regulation of diabetes associated hyperglycemia was proposed by Magnani et al. who successfully encapsulated hexokinase (EC 2.7.1.1), the first enzyme in the glycolytic pathway, responsible for catalyzing the phosphorylation of glucose by ATP to glucose-6-phosphate. *In vitro* studies demonstrated that hexokinase-loaded erythrocytes were able to metabolize twice the amount of extracellular glucose compared to unloaded cells, however, this rate of metabolism was deemed insufficient for maintaining blood glucose within the physiological concentrations [57]. De Flora and co-workers investigated the encapsulation of glucose oxidase (EC 1.1.3.4) isolated from *Aspergillus niger,* another glucose catabolizing enzyme, which converts glucose to hydrogen peroxide and D-glucono-δ-lactone. *In vitro* studies revealed that glucose oxidase-loaded human and mouse erythrocytes were able to consume glucose 3–4 times faster than the native, unloaded erythrocyte. However, there was a reported 10% increase in methemoglobin formation and a several-fold increase in the activity of the pentose phosphate pathway, indicating a glutathione peroxidase-mediated draining of reduced glutathione for removal of hydrogen peroxide that was produced [58,59]. *In vivo* studies in the mouse demonstrated a drastically decreased half-life of loaded cells of one to one and a half hours, compared to 10 days for unloaded erythrocyte carriers, with a selective splenic uptake [59]. On the basis that the encapsulation of hexokinase not only increases the glycolytic rate, but also the production of reducing equivalents (NADPH) in response to an oxidative stress, Rossi et al. proposed the co-encapsulation of hexokinase and glucose oxidase in human and mouse erythrocytes. Using this strategy, the authors were able maintain methemoglobin concentrations within an acceptable range and demonstrate a large increase in glucose consumption [60]. *In vivo* studies in the diabetic mouse model, C57BL/Ks-*db*/*db*/01a, revealed that a single intraperitoneal administration of hexokinase/glucose oxidase-loaded erythrocytes could sustain a near normal blood glucose concentration for seven days, while repeated administrations at 10-day intervals were effective in regulating blood glucose at physiological levels for more than 30 days [60].

#### 3.3.4. Hyperlactatemia

The erythrocyte encapsulation of lactate 2-monooxygenase (EC 1.13.12.4) and L-lactate oxidase (EC 1.1.3.2), two lactate-catabolizing enzymes, as strategies for the treatment for hyperlactatemia was explored by Garin et al. [61]. Lactate 2-mono-oxygenase metabolizes lactate in the presence of oxygen to acetate and carbon dioxide, whereas lactate oxidase metabolizes lactate to pyruvate and hydrogen peroxide. *In vitro* studies in mouse and human erythrocytes demonstrated that encapsulated lactate 2-monooxygenase had a low affinity for lactate, operating at a low rate in the presence of lactate concentrations found in hyperlactatemia (5–20 mM). Erythrocyte encapsulated lactate oxidase however was able to provide a constant catabolic rate under the same range of blood lactate concentrations, but was limited by the generation of hydrogen peroxide which is toxic to the erythrocytes. The co-encapsulation of both enzymes provided a significant rate of lactate metabolism *in vitro*, over a range of 1–30 mM lactate and counteracted the production of hydrogen peroxide by increasing the amount of glucose metabolized in the pentose phosphate pathway. The *in vivo* efficacy of the loaded cells in removing blood lactate in the mouse, however, could not be demonstrated due to the high aerobic capacity and high lactate metabolism of this species [61].

#### 3.3.5. Glucose-6-Phosphate Dehydrogenase Deficiency

Glucose-6-phosphate dehydrogenase (EC 1.1.1.49) deficiency (OMIM # 300908) is caused by mutations in the *G6PD* gene. The enzyme catalyzes nicotinamide adenine dinucleotide phosphate (NADP) to its reduced form, NADPH, in the pentose phosphate pathway. NADPH protects cells from oxidative damage and because mature erythrocytes cannot generate NADPH by any other pathway, a deficiency in glucose-6-phosphate dehydrogenase increases the vulnerability of erythrocytes to oxidative stress. The common clinical manifestations of the disease are neonatal jaundice and acute or chronic hemolytic anemia triggered by exogenous agents [62]. Gerli and co-workers demonstrated that human glucose-6-phosphate dehydrogenase deficient erythrocytes loaded with enzyme extracted from yeast were able to acquire some capacity to reduce glutathione [63]. In the study of Morelli et al., homogeneous human glucose 6-phosphate dehydrogenase was successfully encapsulated into the erythrocytes of individuals affected by glucose 6-phosphate dehydrogenase deficiency, to levels observed in unaffected erythrocytes, with a consequent normalization of the pentose phosphate pathway [64].

#### 3.3.6. Adenosine Deaminase Deficiency

The application of the erythrocyte carrier as an enzyme replacement for adenosine deaminase deficiency (OMIM # 608958) due to severe combined immunodeficiency was explored by Bax et al., [65,66,67,68,69]. The disorder is caused by mutations in the *ADA* gene which encodes for adenosine deaminase (EC 3.5.4.4), the enzyme responsible for catalyzing the irreversible deamination of adenosine and 2’-deoxyadenosine in the purine catabolic pathway. The subsequent accumulation of 2’-deoxyadenosine results in its preferential phosphorylation to deoxyadenosine triphosphate (dATP) which accumulates intracellularly to high levels, resulting in an impairment of lymphocyte differentiation and proliferation through the inhibition of ribonucleotide reductase. The disease is characterized by low levels of immunoglobulins, a virtual absence of T-lymphocytes and markedly reduced levels of B-lymphocytes and natural killer cells [70]. The encapsulation of native adenosine deaminase and the licensed pharmaceutical preparation, polyethylene glycol-conjugated adenosine deaminase (pegademase), within human erythrocyte carriers was investigated. The rationale was to prolong the *in vivo* circulatory half-life of these enzyme preparations and maintain therapeutic blood levels. The efficiencies of native enzyme and pegylated enzyme encapsulation were 50% and 9%, respectively, thus indicating that the polyethylene glycol side-chains were impeding the encapsulation of the pegylated preparation. The biochemical characteristics and the osmotic fragility of loaded cells were not adversely affected by the encapsulation of either enzyme preparation [65,66,68]. *In vivo* survival studies of pegademase-loaded erythrocytes were conducted by labeling with Chromium [^51^Cr] and infusing into an ADA-deficient adult patient and demonstrated a mean cell half-life of 16 days. Erythrocyte encapsulated pegademase and native ADA had *in vivo* circulatory half-lives of 20 and 12.5 days, respectively; these are substantially greater than the plasma half-lives of the unencapsulated preparations, which are less than six days [68]. A clinical evaluation of erythrocyte-encapsulated native adenosine deaminase therapy was conducted over a period of nine years in a single patient who received a total of 225 infusions every 2–3 weeks. Observations included a reduction in the erythrocyte dATP concentration to between 24 and 44 µmol/L, compared to 234 µmol/L on diagnosis and a normalization of serum immunoglobulin levels. Also reported were a reduction in the frequency of respiratory problems and a decline in the forced expiratory volume in one second and vital capacity reduced compared with the four years preceding erythrocyte therapy [69].

#### 3.3.7. Mitochondrial Neurogastrointestinal Encephalomyopathy (MNGIE)

MNGIE (OMIM # 603041) is an ultra-rare, fatal disorder caused by mutations in the nuclear *TYMP* gene which leads to a deficiency in thymidine phosphorylase activity (EC 2.4.2.4). Thymidine phosphorylase plays a pivotal role in the deoxyribonucleoside salvage metabolic pathway through catalyzing the reversible phosphorylation of thymidine and 2’-deoxyuridine to 2-deoxyribose 1-phosphate and their respective bases, thymine and uracil. In the absence of thymidine phosphorylase activity there is a systemic accumulation of thymidine and 2’-deoxyuridine which generate imbalances within the mitochondrial deoxyribonucleotide pools, causing mitochondrial DNA (mtDNA) point mutations, depletion and deletion abnormalities—and ultimately mitochondrial dysfunction. The disorder is characterized by gastrointestinal dysmotility, cachexia, peripheral neuropathy, ophthalmoplegia, ptosis and leukoencephalopathy [71]. Although there are no licensed therapies for patients with MNGIE, there are a number of experiment treatment approaches under investigation and this is attributed to MNGIE being one of the few mitochondrial disorders where the molecular abnormality is metabolically and physically accessible to manipulation. Erythrocyte encapsulated thymidine phosphorylase (EE-TP) is one of these experimental approaches, where recombinant *E. coli* enzyme was employed as the source of enzyme [71,72,73,74]. In the first proof of concept study, one dose of EE-TP (1020 units encapsulated within 20.25 × 10^10^ erythrocytes) was administered to a patient with MNGI.E. Three days post infusion, there was a reported decrease in the urinary excretion of thymidine and 2’-deoxyuridine, to 6% and 13%, respectively of the amounts excreted pre-therapy. Plasma metabolites concentrations were also shown to decrease in parallel [75]. Pre-clinical studies conducted in the mouse and dog revealed no safety issues that would preclude a clinical study of EE-TP, other than the production of anti-thymidine phosphorylase antibodies in a small proportion of animals [76,77]. Based on these encouraging results, a compassionate use program was conducted in four further patients, with participants receiving treatment in accordance with the provisions of Schedule 1 of the Medicines for Human Use (marketing authorizations, etc.) Regulations SI 1994/3144, where schedule 1 provides an exemption from the need for a marketing authorization for a relevant medicinal product, which is supplied on an individual patient basis to fulfil a "special need". A chromium [^51^Cr]–labeling study of EE-TP was conducted in one patient to evaluate the *in vivo* survival characteristics of the patient’s enzyme-loaded erythrocytes and this revealed a normal circulating mean cell life and half-life of 108 and 32 days, respectively [78]. The administration of doses ranging between 3.7 to 108 U/kg body weight/4 weeks indicated that doses greater than 47 U/kg/4 weeks resulted a greater reduction or an elimination of plasma and urine metabolites. One patient was reported to gain four kg in weight, three months after initiating therapy and this coincided with a decrease in nausea and vomiting, increased ability to walk longer distances and an increase in the physical and mental components of the SF36 health and well-being survey. A second patient showed significant clinical improvements in bilateral muscle power, gait and balance, sensory ataxia and fine finger function, after 23 months of EE-TP therapy. In addition, a weight gain of 5.8 kg was reported and a fall in plasma creatine kinase activity from 1200 U/L pre-therapy to levels within the normal reference range. Patient reported outcomes included an ability to walk longer distances, climb stairs without assistance, tie shoe laces, feel the sensation of sand, and a return to weight training and guitar playing [78,79,80]. Adverse events for EE-TP that were recorded in this compassionate study included nausea, coughing spasms and erythema of the face and neck in two of the four patients. These were transient in nature, occurring within the first five minutes of EE-TP infusion and did not lead to patient discontinuation of EE-TP therapy. Pre-medication with antihistamine, corticosteroid anti-inflammatory and anti-emetic drugs provided a strategy for successfully managing these reactions [78,79]. Specific antibodies to thymidine phosphorylase were detected in one patient, after the tenth treatment cycle. However, this did not raise any concerns with regard to neutralizing antibodies as the efficacy of encapsulated thymidine phosphorylase in metabolizing the plasma metabolites improved over the 5.5 years of EE-TP administration to this patient, and positive clinical responses to treatment continued to be recorded [81]. The development of specific antibodies is not surprising considering that senescent erythrocytes are naturally sequestered from the vascular compartment by macrophages of the monocyte–macrophage system, where macrophages are able to present peptides to T-lymphocytes [82]. EE-TP has received MHRA clearance for a Phase II, multi-center, multiple-dose, open-label trial without a control to determine the safety, tolerability, pharmacodynamics and efficacy this treatment approach in patients with MNGIE (https://clinicaltrials.gov/ct2/show/NCT03866954) [83]. EE-TP has Orphan designation by the EMA and FDA.

#### 3.3.8. Hyperuricemia

Due to the loss of uricase (EC 1.7.3.3) activity during the evolution of hominids, uric acid is the end product of purine metabolism in humans and other higher primates and is excreted in the urine. Hyperuricemia is caused by abnormally elevated concentrations of blood uric acid and arises through several mechanisms, including reduced renal excretion or increased uric acid production. The concept of utilizing the erythrocyte as a mechanism for degrading extracellular molecules was investigated by Ihler et al. through examining the *in vitro* metabolism of extracellular uric acid by human erythrocytes-loaded with uricase. Uric acid was shown to be transported into enzyme-loaded cells at the same rate as in native erythrocytes, however, the generation of hydrogen peroxide through the metabolism of uric acid to allantoin raised the concern of inducing oxidative stress damage [84]. To address this, Magnani et al. coupled uricase to the extracellular human erythrocyte membrane by a biotin–avidin–biotin–enzyme bridge, and in doing so, demonstrated a more efficient *in vitro* metabolism of uric acid, compared to that by encapsulated enzyme [85].

#### 3.3.9. Phenylketonuria

Phenylketonuria (OMIM # 261600) is an autosomal recessive disorder caused by mutations in the *PAH* gene, leading to a deficiency in the enzyme responsible for the hydroxylation of phenylalanine to tyrosine, phenylalanine hydroxylase (EC 1.14.16.1). If untreated, phenylketonuria can result in impaired postnatal cognitive development resulting from a neurotoxic effect of hyperphenylalaninemia [86]. Another enzyme capable of metabolizing phenylalanine is phenylalanine ammonia lyase (EC 4.3.1.24). This enzyme is not expressed in mammals, but is expressed in plants, some bacteria, yeast and fungi and catalyzes the conversion of phenylalanine to ammonia and trans-cinnamic acid. Its potential application via erythrocyte encapsulation as a therapy for phenylketonuria was first proposed by Sprandel and Zoller, with a short-term *in vivo* study in mice demonstrating the effectiveness of this strategy [87]. Erythrocytes loaded with recombinant *Chromobacterium violaceum* phenylalanine hydroxylase, together with the cofactor tetrahydrobiopterin, were shown to reduced blood phenylalanine levels when administered to wild-type mice, but not when administered to the BTBR Pah*^enu2^* phenylketonuria mouse model [88]. In contrast, the repeated administration of erythrocytes loaded with recombinant *Anabaena variabilis* phenylalanine ammonia lyase (rAvPAL) at 9–10-day intervals for 10 weeks, was able to provide a sustained reduction in blood phenylalanine levels in the BTBR Pah*^enu2^* mouse model and also reverse fur pigmentation. Although, anti-enzyme antibodies were generated, these did not affect the efficacy of the encapsulated enzyme [89]. More recent studies examining the clinical efficacy of erythrocyte-loaded rAvPAL in the early treatment of the BTBR Pah*^enu2^* mouse demonstrated a normalized of blood and brain phenylalanine concentrations and prevented cognitive developmental failure, depletion of brain serotonin, dendritic spine abnormalities and myelin basic protein reduction [90]. Rubius Therapeutics, Inc received FDA approval for a Phase Ib clinical trial of genetically engineered erythrocytes expressing phenylalanine ammonia lyase (RTX-134). This approach involves the genetic engineering of CD34^+^ cells, collected from a healthy O negative donor, with a lentiviral vector to express enzyme phenylalanine ammonia lyase within the cells. The cells are expanded and differentiated until the nucleus is ejected, resulting in the mature reticulocyte. The trial is reported to be active, but no longer recruiting, and is designed to determine a preliminary dose and to inform on a dosing schedule that is deemed safe, tolerable and potentially effective (https://clinicaltrials.gov/ct2/show/NCT04110496). Recent media coverage has, however reported that the trial will be discontinued due to results from the first patient being uninterpretable. This is thought to be due to the low dose of cells administered and the lack of sensitivity of the flow cytometry assay used to detect circulating cells.

### 3.4. Antitumor Therapy

The implementation of the erythrocyte carrier in antitumor therapy has focused on the use of encapsulated enzymes and represents one of the most clinically advanced applications is for the treatment of acute lymphoblastic leukemia (ALL). This strategy is based on tumor cells not being able to synthesize adequate amounts of amino acids and therefore depend on extracellular sources. The administration of specific enzymes enables the depletion of plasma amino acids, thus starving the tumor cells of amino acids necessary for DNA, RNA and protein synthesis and leading ultimately to cell death. The antitumor characteristics of L-asparaginase (EC 3.5.1.1) was first investigated by Kidd, who reported lymphoma regression in mice and rats in response to guinea pig plasma [91]. In the early 1970s, native *Erwinia chrysanthemi* and *E. coli*-derived l-asparaginase were introduced into chemotherapy protocols for the treatment of ALL. However, its use was associated with toxicity against the liver and pancreas. High immunogenicity, manifesting as hypersensitivity reactions and/or the neutralization of asparaginase activity without any signs of hypersensitivity were also observed. The enzymes had a short circulatory half-life, ranging between 8 and 30 h thus necessitating frequent infusions [92]. Pegylated formulations of *E. coli*
l-asparaginase were developed to overcome these limitations, however these preparations are associated with hepatotoxicity, pancreatitis and thrombosis [93]. Updike et al. were the first to employ the erythrocyte carrier as a strategy to counteract the problems of antigenicity and host proteolytic degradation of asparaginase. *In vivo* studies demonstrated that circulating asparaginase activity in monkeys injected with enzyme-loaded erythrocytes was two orders of magnitude higher than that in animals injected with native asparaginase [94,95]. In addition, monkeys that received a single injection of erythrocyte-loaded enzyme were able to suppress plasma asparagine concentrations for 20 days compared to 10 days for animals that received the native enzyme [96]. Further preliminary *in vivo* studies in the mouse and human demonstrated the safety of asparaginase-loaded erythrocytes and their efficacy in reducing plasma asparagine concentrations [96,97,98,99,100].

Erytech Pharma have developed a number of programs to further investigate the safety and efficacy of erythrocyte encapsulated asparaginase for the treatment of several types of cancers. In their Phase I/II study of *E. coli* erythrocyte-encapsulated l-asparaginase (GRASPA) conducted in adults and children with ALL, they were able to demonstrate an effective depletion of L-asparagine. A single injection of the highest dose (150 IU/kg) was shown to provide similar results to eight intravenous injections of 10,000 IU/m^2^ of the native enzyme. In addition, compared to the native enzyme, GRASPA exhibited a reduction in the number and severity of allergic reactions and a trend towards less coagulation disorders [101]. A subsequent Phase II trial evaluating the safety and efficacy of GRASPA was conducted in patients 55 years of age and older with Philadelphia chromosome-negative ALL. The primary efficacy endpoint of asparagine depletion <2 µmol/L for at least seven days was achieved in 85% and 71% of patients who received 100 and 150 IU/kg, respectively, but not with those patients who received 50 IU/kg. Although no dose limiting toxicity was observed at the lowest dosage, in the 100 IU/kg and 150 IU/kg cohorts, toxicities were 15% and 36%, respectively. This suggested that the 100 IU/kg dose had the best safety/efficacy profile for these elderly patients [102]. Erythrocyte encapsulated asparaginase was subsequently extended to a Phase I clinical trial for the treatment of patients with pancreatic adenocarcinoma with null/low asparagine synthetase expression. The intervention (ERY–ASP) was well tolerated by patients, and demonstrated no dose limiting toxicities reported [103]. A follow-on Phase IIb study evaluating erythrocyte-encapsulated asparaginase (relabeled as eryaspase) in combination with chemotherapy in second-line advanced pancreatic adenocarcinoma was conducted. This demonstrated significantly prolonged overall survival and progression free survival compared with chemotherapy alone, with a 40% reduction in the risk of death on average over time. The therapy combination was shown to be generally well tolerated and revealed no unexpected safety findings [104]. A Phase III study in patients with ductal adenocarcinoma of the pancreas who have failed one prior line of systemic anticancer therapy for advanced pancreatic cancer and have measurable disease is currently active and recruiting. The primary outcome measure is whether the addition of eryaspase to chemotherapy improves overall survival when compared to chemotherapy alone (https://clinicaltrials.gov/ct2/show/NCT03665441). Following the positive clinical data obtained from the pancreatic cancer studies, Erytech are now expanding their solid tumor studies to include selected metastatic triple-negative breast cancer. The ongoing Phase II/III trial is evaluating eryaspase in combination with gemcitabine and carboplatin chemotherapy, compared to chemotherapy alone as a first-line treatment (https://clinicaltrials.gov/ct2/show/NCT03674242).

Other enzymes, with relevance to tumor starvation which have been encapsulated include methionine-γ-lyase (EC 4.4.1.11) and arginine deiminase (EC 3.5.3.6). The administration of erythrocyte encapsulated *Pseudomonas putida* methionine-γ-lyase to mice grafted with human gastric adenocarcinoma and glioblastoma was effective in providing a sustained and significant plasma methionine depletion *in vivo* and a reduction in tumor growth [105]. A significant inhibition of tumor growth was observed in mice bearing orthotopic EMT-6 syngeneic breast carcinoma when infused with erythrocyte-encapsulated methionine-γ-lyase at a dose of 60 U/kg and in combination with anti-mouse PD-1 antibody, compared to the separate entities. Survival time was 35 days for the combined therapy, compared to 23 days for the separate therapies [106]. With regard to erythrocyte encapsulated arginine deiminase, a dose of 10.4 IU/mL administered to CD1 mice was shown to completely deplete plasma arginine over a period of five days compared to the same dose of free enzyme. The depletion was sustained for 24 h, with the plasma levels returning to baseline by 2.5 days [107].

## 4. Challenges and Limitations of Erythrocyte-Based Enzyme Therapy

The limitation of the erythrocyte enzyme carrier is that its metabolic capacity is restricted to the vascular compartment and monocyte–macrophage system. For the former, a further restriction for the encapsulated enzyme, would be the inability of the enzyme’s substrate to permeate the erythrocyte membrane. Thus, the spectrum of clinical applications to which the erythrocyte enzyme carrier can be applied to is fairly narrow. As with most cell-based therapies, the translation of the erythrocyte carrier into the clinical setting faces many complex process-related and regulatory challenges. The enzyme to be encapsulated or conjugated to the erythrocyte membrane will require manufacture in compliance with current good manufacturing practice (cGMP). Although advancements in the field of recombinant DNA biotechnologies have facilitated the manufacture of therapeutic enzymes, many challenging activities exist, including full characterization of cell banks, process- and product-related impurities and enzyme; validation of analytical procedures; formulation design; and stability studies. The transition of the manufacture of the erythrocyte carrier from a laboratory-based scale, to a scale that is clinically relevant is a critical step in the development pathway. Manufacture can be via a decentralized automated process, such as the Red Cell Loader as employed by Erydel or via a centralized process as used by Erytech Pharma [108,109]. Either way, the process will need to be performed in a closed-circuit system, using single use, sterile disposables. A key regulatory requirement is the manufacture of a reproducible product. Standardizing the loading procedure (and thus dose) when autologous erythrocytes are employed can be difficult to achieve, and may necessitate the consideration of individualized metrics such as patient hematocrit or red blood cell count in the manufacturing process. To minimize substantial differences in the quality of the end product and to produce a consistent encapsulation rate, Erytech Pharma who employ homologous donated erythrocytes, adjust the process operating conditions such as erythrocyte flow rate and osmolality of buffers according to the osmotic fragility of the initial donor erythrocyte pellet [109]. Prior to clinical batch release by a pharmacist, the product will need to conform to product specifications. For erythrocyte carriers manufactured from homologous blood, the 72-h shelf-life specified by Erytech provides sufficient time to conduct the necessary tests [109]. However, erythrocyte carriers manufactured from blood collected from autologous blood require a much shorter shelf-life (30 min) thus providing a major hurdle to testing. The MHRA approved Phase II trial of EE-TP was able to navigate this issue by proposing a retrospective testing of two of the release parameters. The impact of manufacturing process and chemical modification (where enzymes are conjugated to the erythrocyte membrane) on the function, viability and biocompatibility of the erythrocyte will also need to be established to ensure safety of the medicinal product.

## 5. Conclusions

The erythrocyte carrier has the advantage of counteracting some of the key issues associated with the administration of free enzymes by virtual of shielding the enzyme from the immune system and plasma proteases, and thereby enhance the circulatory half-life and minimize immunogenic reactions. The feasibility of erythrocyte-mediated enzyme therapy to deplete pathological molecules in the circulation has been investigated by many workers using animal models and humans. This paper provides an overview of the immense range of applications that have been investigated, covering the early studies of the 1970s up to the more recent biomedical applications that have overcome the process and regulatory challenges that face cell therapy development and have reached the clinic; these are summarized in Table 1. Although there are currently no licensed erythrocyte-mediated enzyme therapeutics, this is very likely to change over the next few years, with several products undergoing advanced clinical development for conditions where there are unmet needs.

## Figures and Tables

**Figure 1 pharmaceutics-12-00435-f001:**
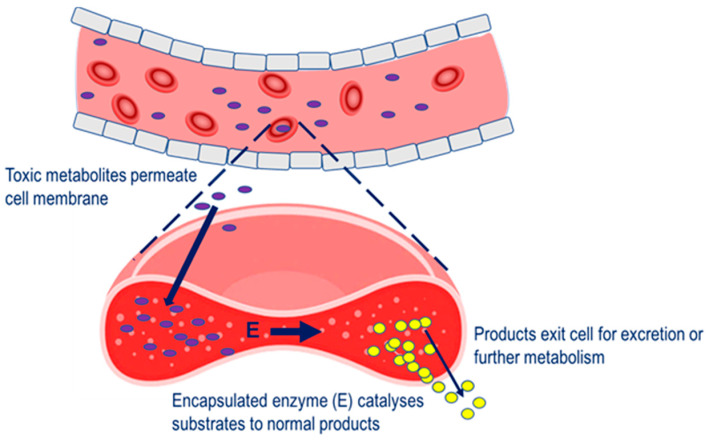
Strategy for the removal of pathological metabolites from the circulation. Pathological metabolites in the blood enter the erythrocyte carrier, to undergo metabolism by the encapsulated enzyme (E).

**Figure 2 pharmaceutics-12-00435-f002:**
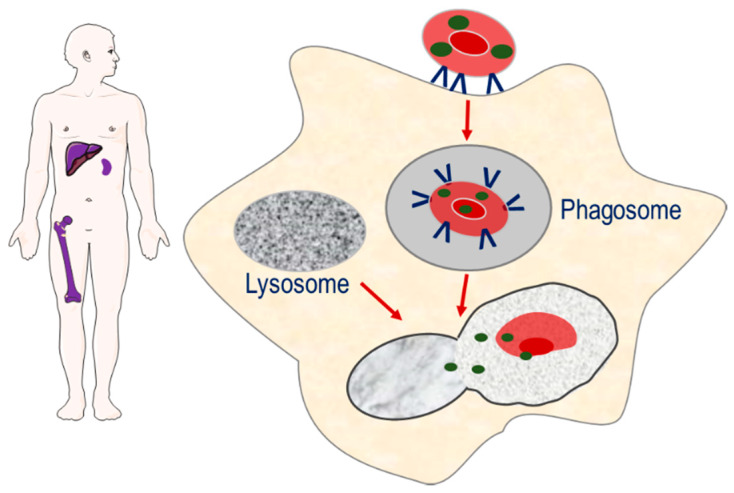
Strategy for targeting enzymes to the monocyte–macrophage system of the spleen, liver and bone marrow. Senescent erythrocyte carriers loaded with enzymes are removed from the circulation by macrophages. The resulting phagosome fuses with macromolecule-laden lysosomes, resulting in the release of enzyme and breakdown of macromolecules.

**Figure 3 pharmaceutics-12-00435-f003:**
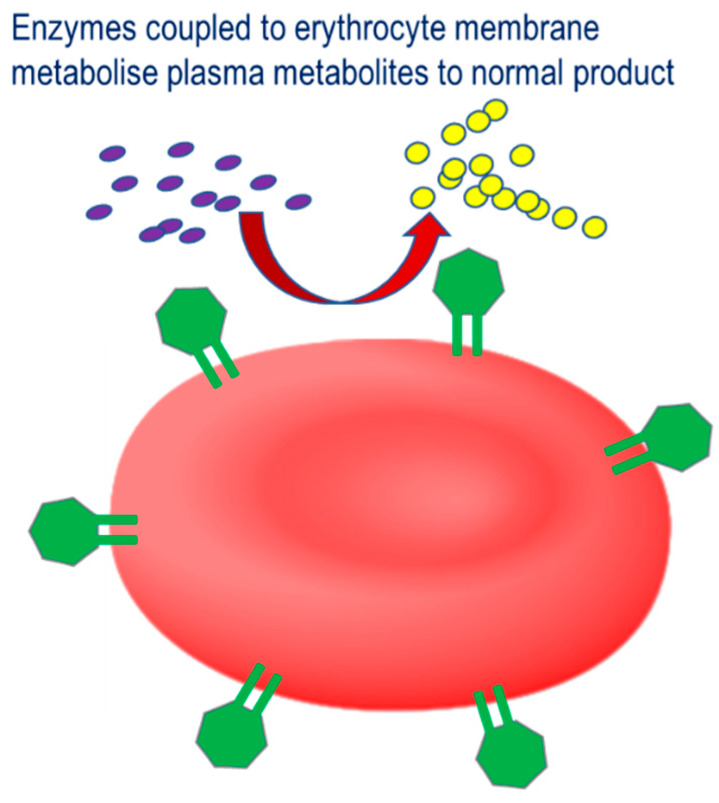
Therapeutic strategy for therapeutic enzymes coupled to the erythrocyte membrane. Pathological plasma metabolites are catalyzed by the conjugated enzyme to their corresponding non-pathological product.

**Figure 4 pharmaceutics-12-00435-f004:**
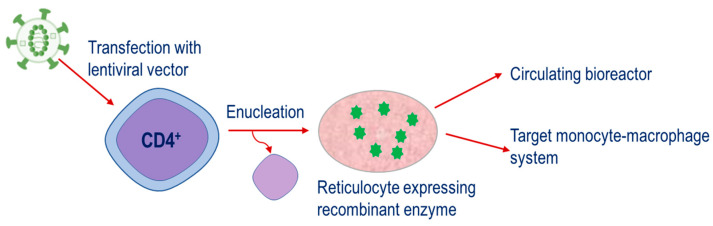
Strategy for therapeutic enzymes produced through the transfection of CD34^+^ cells with lentiviral vectors containing constructs encoding for the enzyme of interest. Cells are expanded and differentiated until the nucleus is ejected, resulting in the mature reticulocyte expressing the recombinant enzyme.

**Table 1 pharmaceutics-12-00435-t001:** Summary of *in vitro*, *in vivo* and clinical applications of erythrocyte-mediated enzyme therapy.

Therapeutic Application	Therapeutic Target/Disorder	Encapsulated/Conjugated Enzyme	Investigations
Detoxification of exogenous chemicals	Cyanide [8,9,10,11,12,13]	Rhodanase	Mouse *in vivo*
	Paraoxon [14,15]	Phosphotriesterase	Mouse *in vivo*
		Paraoxonase	Mouse *in vivo*
	Ethanol [16,17,18,19,20]	Acetaldehyde dehydrogenase	Mouse *in vivo*
		Alcohol dehydrogenase/aldehyde dehydrogenase	Mouse *in vitro*/*in vivo* Human *in silico*/*in vitro*
	Methanol [21,22]	Alcohol oxidase	Mouse *in vivo* Mouse *in vivo*
		Formate dehydrogenase	
	Lead [23,24,25]	δ-aminolevulinic acid dehydratase	Mouse *in vitro*/*in vivo* Human clinical
	Hydrogen gas [26]	Hydrogenase	Human *in vitro*
Thrombolytic therapy	Plasminogen [27,28,29,30,31]	Urokinase	Human *in vitro* Mouse *in vivo*
		Streptokinase	Human *in vitro* Mouse *in vivo*
		tPA	Human *in vitro* Mouse/rat *in vivo*
		Brinase	Rabbit *in vitro*
Treatment of metabolic disease	Sphingolipids (Lysosomal storage disorders) [32,33,34,35,36,37]	β-glucosidase	Human *in vitro*
		β-galactosidase	Human *in vitro*
		β-glucuronidase	Mouse *in vivo*
		β-glucocerebrosidase	Human *in vitro*/clinical
		Alglucerase	Human *in vitro*
	Ammonia (hyperammonemia) [38,39,40,41,42]	l-glutamate dehydrogenase	Human *in vitro* Mouse *in vivo*
		Glutamine synthetase	Mouse *in vivo*
		Glutamate dehydrogenase/alanine aminotransferase	*In silico* Human *in vitro* Mouse *in vivo*
	Arginine and ammonia (arginase-1 deficiency) [43]	Arginase	Human *in vitro*
	Urea and ammonia (chronic renal failure) [44,45,46,47]	Urease/alanine dehydrogenase	Human *in vitro* Sheep *in vivo*
	Glucose (hyperglycemia) [48,49,50,51]	Hexokinase	Human *in vitro*
		Glucose oxidase	Human *in vitro* Mouse *in vitro*/*in vivo*
		Hexokinase/glucose oxidase	Human *in vitro* Mouse *in vitro*/*in vivo*
	Lactate (hyperlactatemia) [52]	Lactate 2-mono-oxygenase	Mouse/human *in vitro*
		Lactate oxidase	Mouse/human *in vitro*
		Lactate 2-mono-oxygenase/ lactate oxidase	Mouse *in vitro*/*in vivo* Human *in vitro*
	NADP (Glucose-6-phosphate dehydrogenase deficiency) [3,54]	Glucose-6-phosphate dehydrogenase	Human *in vitro*
	Adenosine and 2‘-deoxyadenosine (Adenosine deaminase deficiency) [55,56,57,58,59]	Adenosine deaminase Pegademase	Human *in vitro*/*in vivo* Human *in vitro*/clinical
	Thymidine and 2’-deoxyuridine (MNGIE) [60,61,62,63,64,65,66,67,68,69,70,71,72]	Thymidine phosphorylase	Mouse/dog *in vivo* Human Phase II
	Uric acid (hyperuricemia) [73,74]	Uricase	Human *in vitro* Mouse *in vivo*
	Phenylalanine (phenylketonuria) [75,76,77,78]	Phenylalanine ammonia lyase	Mouse *in vitro*/*in vivo*
		Phenylalanine hydroxylase	Mouse *in vivo*
		RTX-134	Human Phase Ib
Antitumor therapy	Asparagine (acute lymphoblastic leukemia) [79,80,81,82,83,84,85,86,87,88,89,90]	Asparaginase GRASPA	Monkey *in vivo* Human Phase I/II
	Asparagine (pancreatic adenocarcinoma) [91,92]	ERY-ASP Eryaspase	Human Phase I Human Phase IIb/ Phase III
	(triple negative breast cancer)	Eryaspase	Human Phase II/III
	Methionine (gastric adenocarcinoma, glioblastoma, breast carcinoma) [93,94]	Methionine-γ-lyase	Mouse *in vivo*
	Arginine [95]	Arginine deiminase	Mouse *in vivo*

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
