# Peer review of "Erythrocytes as Carriers of Therapeutic Enzymes"

_pharmaceutics, 2020, doi:10.3390/pharmaceutics12050435_

Round 1
Reviewer 1 Report
Please see the attachment.

Author Response
The author thanks the reviewer for their constructive feedback. Please find below the author’s responses to the queries raised:
1.1 Many long sentences were used in the paper. But they were not used correctly. In abstract, the sentence in line 11-13 should be expressed smoothly.
This sentence now reads as: However, the metabolic and clinical efficacies of parenterally or intramuscularly administered therapeutic enzymes are very often limited by short circulatory half-lives, hypersensitive and immunogenic reactions.
1.2 “Over the past five decades” in line 13 and line 140 is not accurate. The similar review and research can be found in 1985 or earlier.
Could the reviewer please clarify this request, as my understanding is the erythrocyte has been investigated as a carrier of therapeutic agents since the 1970s? This is five decades (50 years). Many thanks.
1.3 In line18, the front mentioned 4 keywords should revised to 2 keywords. “drug delivery” and “targeted drug delivery” are repeated.
Targeted drug delivery has been removed. I am not sure if I understand what is meant here as the instructions stated that we could “ List three to ten pertinent keywords specific to the article”.
1.4 Perhaps therapeutic strategies and therapeutic applications should be described briefly in the abstract.
The following has been added to the Abstract (Lines 16-22): These strategies include their application as circulating bioreactors, targeting the monocyte-macrophage system, the coupling of enzymes to the surface of the erythrocyte and the engineering of CD34+ hematopoietic precursor cells for the expression of therapeutic enzymes. An overview of the diverse biomedical applications for which they have been investigated is also provided, including the detoxification of exogenous chemicals, thrombolytic therapy, enzyme replacement therapy for metabolic diseases and anti-tumour therapy.
1.5 In line 42, it should be indicated in which species the half-life can reach 19-29 days.
The following has been added to Line 49: in the human
1.6 The phrase “blood-based diffusible molecules” in line 51 is not expressed clearly, since there has no explanation for what is blood-based.
The sentence has been amended as follows (Line 57-59): In the first, the loaded erythrocytes can be employed as circulating bioreactors whereby blood-based molecules diffuse into the erythrocyte and are degraded.
1.7 “3.3.7. Mitochondrial neurogastrointestinal encephalomyopathy (MNGIE)”, the ”half-life of 108 and 32 days respectively” in line 487 should point out what do the two half-lives refer to.
The position of commas has been amended to make this clearer. (Line 521).
1.8 The titles of 3.3.1-3.3.9 perhaps should be expressed as ”Enzyme replacement for ----”.
In order to make the manuscript more succinct, the following sentence has been added to make it clear to the reader that a number of metabolic disorders will be discussed (see Lines 281-283): The following provides a discussion of the metabolic disorders to which the erythrocyte carrier has been applied to.
1.9 In abstract and “3. Therapeutic application”, classification basis should be described for the applied diseases.
The following has been added to the Abstract (Line 19-22): An overview of the diverse biomedical applications for which they have been investigated is also provided, including the detoxification of exogenous chemicals, thrombolytic therapy, enzyme replacement therapy for metabolic diseases and anti-tumour therapy.
1.10 The sentences in line:
23-26. Has replaced with “have”. See Line 28
29-31. This sentence has been divided (Lines 34-37) : However, despite these strategies, the metabolic and clinical efficacy of parenterally or intramuscularly administered therapeutic enzymes is still limited. This is principally due to short circulatory half-lives, hypersensitivity reactions and immunogenicity.
34-36. “To” has been added to this sentence. See Line 41
39-41 ” and “The erythrocyte carrier” : This sentence has been divided into two (lines 45-48): The erythrocyte carrier has been extensively studied as a strategy for over-coming these limitations and increasing therapeutic efficacy. For a majority of the therapeutic applications investigated, the ability of the cell to reseal after creating pores in the membrane has been exploited for the purpose of introducing therapeutic agents[3–5]
25-127. Sentence has been divided (Lines 133-135): The fourth strategy is a relatively new approach. This is based on the engineering of CD34+ hematopoietic precursor cells to express therapeutic enzymes and then their subsequent differentiation until the nucleus is ejected, resulting in the mature reticulocyte.
155-158. The sentence has been amended to (Lines 165-167): Sodium thiosulphate does not readily permeate cell membranes and therefore is not able to distribute to sites of thiosulfate sulfotransferase or cyanide localization; this provided the rationale for investigating the erythrocyte carrier as an alternative approach to cyanide antagonism.
165-166. This sentence has been amended to (Lines 180-181): However, neither of these antidotes are able to degrade parathion.
183-184, 184-186. This sentence has been divided (Lines 199-202): The application of the erythrocyte carrier as an alcohol detoxifier was first proposed by Magnani et al. They demonstrated that mice administered acetaldehyde dehydrogenase-loaded erythrocytes intraperitoneally had 35% less blood acetaldehyde compared to controls, one hour after receiving an acute dose of ethanol[16]
193-196. This sentence has been divided (Lines 209-2013): By employing a mathematical modelling approach and then conducting a subsequent in vitro study, Alexandrovich et al. were able to theorise and then demonstrate the rate limiting step of external ethanol oxidation. They found this was due to the rate of nicotinamide-adenine dinucleotide (NAD+) generation in erythrocyte glycolysis, rather than the activities of the loaded enzymes.
200-203. This sentence has been amended to (Lines 2016-220): Formic acid metabolism is mediated through a tetrahydrofolate-dependent pathway by folate-dependent enzymes. Humans have 60% less liver folate concentrations compared to mice and rats, and for this reason humans are more sensitive to methanol poisoning.
251-253. This sentence has been amended to (Lines 270-273): Free tPA was able to lyse pulmonary clots that had lodged before administration, but not those that lodged after injection. Erythrocyte conjugated- tPA, however was more selective in lysing nascent over preexisting pulmonary emboli and arterial clots, an effect that is most likely a result of restricting the diffusion of tPA into fibrin [30].
272-273. This has been clarified (Lines 293-299): Deloach and coworkers demonstrated that human erythrocytes-loaded with E.coli β-galactosidase could be phagocytosed by bone marrow macrophages that had been matured in vitro. Electron microscopy revealed that intact or partly degraded erythrocytes were localised in intracellular vacuoles, which were presumed to be phagolysosomes. The activity of the untaken β-galactosidase disappeared with a half-life of 15-30 hours, which was consistent with the enzyme being degraded within the lysosomes [32].
279-282. The gene is now italicised (Line 306): GBA
291-294. This sentence has been amended to (317-320): Studies investigating the in vitro uptake of glucocerebrosidase by Gaucher patient monocytes demonstrated that enzyme loaded erythrocytes coated with human IgG and agglutinated with anti-human serum were more avidly phagocytosed compared to uncoated loaded erythrocytes.
322-324. This sentence has been amended to (Lines 351-354): After 24 hours of incubation, the ammonia concentration was 15 µmol/L, this presenting 15% of the zero time concentration and therefore providing support for this approach [38]
324-327. This sentence has been divided (Lines 354-361): Subsequent in vivo studies were performed in mice where hyperammonaemia was induced through the intraperitoneal injection of 33 U of urease/Kg of body weight. One hour after urease administration the blood concentration of ammonia was 1333 µmol/L and this declined to 135 µmol/l by 25 hours. By administering glutamate dehydrogenase-loaded erythrocytes to these mice, the blood concentration of ammonia could be reduced by 50%, 2 hours after urease injection. The ammonia concentration remained unchanged in control mice that received native washed erythrocytes[39].
339-343.This sentence has been divided (Lines 371-375): To compensate the effect of this, the authors proposed the co-encapsulation of glutamate dehydrogenase and alanine aminotransferase (EC 2.6.1.2). This approached created a metabolic pathway where glutamic acid produced by the glutamate dehydrogenase reaction was converted into α-ketoglutarate by alanine aminotransferase, with α-ketoglutarate then entering the glutamate dehydrogenase reaction.
358-260. This sentenced has been amended to (Lines 390-392): They demonstrated that when the enzyme-loaded erythrocytes were incubated in plasma containing 45 to 375 µmol/L arginine, the extracellular concentration of arginine was reduced by 34 to 53% after one hour [43].
405-409. This sentence has been divided (Lines 437-442): In vitro studies in mouse and human erythrocytes demonstrated that encapsulated lactate monooxygenase had a low affinity for lactate, operating at a low rate in the presence of lactate concentrations found in hyperlactataemia (5‐20 mM). Erythrocyte encapsulated lactate oxidase however was able to provide a constant catabolic rate under the same range of blood lactate concentrations, but was limited by the generation of hydrogen peroxide which is toxic to the erythrocytes.
418-422. This sentence has been divided (Lines 437-442): In vitro studies in mouse and human erythrocytes demonstrated that encapsulated lactate 2‐monooxygenase had a low affinity for lactate, operating at a low rate in the presence of lactate concentrations found in hyperlactataemia (5‐20 mM). Erythrocyte encapsulated lactate oxidase however was able to provide a constant catabolic rate under the same range of blood lactate concentrations, but was limited by the generation of hydrogen peroxide which is toxic to the erythrocytes.
432-434. Could the reviewer please clarify the problem with this sentence?
439-442. This sentence has been amended to (Lines 471-475): The encapsulation of native adenosine deaminase and the licenced pharmaceutical preparation, polyethylene glycol-conjugated adenosine deaminase (pegademase), within human erythrocyte carriers was investigated. The rationale was to prolong the in vivo circulatory half-life of these enzyme preparations and maintain therapeutic blood levels.
443-445. This sentence has been amended to (Lines 475-477): The efficiencies of native enzyme and pegylated enzyme encapsulation were 50% and 9%, respectively, thus indicating that the polyethylene glycol side-chains were impeding the encapsulation of the pegylated preparation.
449-451. This sentence has been amended to (Lines 481-485): Erythrocyte encapsulated pegademase and native ADA had in vivo circulatory half-lives of 20 and 12.5 days, respectively; these are substantially greater than the plasma half-lives of the unencapsulated preparations, which are less than 6 days [58].
453-457. This sentence has been amended to (Lines 487-492): Observations included a reduction in the erythrocyte dATP concentration to between 24 and 44 µmol/L, compared to 234µmol/L on diagnosis and a normalisation of serum immunoglobulin levels. Also reported were a reduction in the frequency of respiratory problems and a decline in the forced expiratory volume in one second and vital capacity reduced compared with the 4 years preceding erythrocyte therapy[59].
529-531. This sentence has been amended to (Lines 564-566): Phenylketonuria (OMIM #261600) is an autosomal recessive disorder caused by mutations in the PAH gene, leading to a deficiency in the enzyme responsible for the hydroxylation of phenylalanine to tyrosine, phenylalanine hydroxylase (EC 1.14.16.1),
529-534. This sentence has been amended to (Lines 568-571): Another enzyme capable of metabolizing phenylalanine is phenylalanine ammonia lyase (EC 4.3.1.24). This enzyme is not expressed in mammals but is expressed in plants, some bacteria, yeast, and fungi and catalyses the conversion of phenylalanine to ammonia and trans-cinnamic acid.
629-630. This sentence has been amended to (Lines 712-713): The feasibility of erythrocyte-mediated enzyme therapy to deplete pathological molecules in the circulation has been investigated by many workers using animal models and humans.
1.11 The content in “3.4. Anti-tumour therapy” should be refined.
Sentences in this section have been shortened and/or amended.
2. Some appropriate and adequate references are mentioned in the review, but some statements should have more related references.
References 10, 15, 16, 24, 27, 32, 41, 51, 62, 70 and 71 have been added to the review.
2.1 In line 166, there may should have some references to support the short action of free paraoxonase.
This statement has been removed as in hindsight, the positioning is not correct.
2.2 In line 223, dose the context of “biochemical decompression” belongs to the part of “Detoxification of exogenous chemicals” ?
This has been left in this section as it refers to the removal of exogenously administered hydrogen gas. This can be toxic for drivers on returning to the surface.
2.3 At the end of part “3.3.1. Lysosomal storage disorders” in line301-303,the author may should have some explanation or references for why this approach has not been advanced in the clinical setting.
The following sentence has been added (lines 336-339): The availability of a number of approved and effective enzyme replacement therapies for Gaucher disease that can be administered in the patient’s home may have contributed to the lack of incentive in developing the erythrocyte carrier further for this disease.
2.4 In line 548, the referred clinical trial has been terminated, it should provide relevant information about the state of the clinical.
The sentence has been amended as follows (Lines 597-603) : The trial is reported to be active but no longer recruiting, and is designed to determine a preliminary dose and to inform on a dosing schedule that is deemed safe, tolerable, and potentially effective (https://clinicaltrials.gov/ct2/show/NCT04110496). Recent media coverage has, however reported that the trial will be discontinued due to results from the first patient being uninterpretable. This is thought to be due to the low dose of cells administered and the lack of sensitivity of the flow cytometry assay used to detect circulating cells.
And the long sentence in line 544-550 should revised, expressed smoothly and in correct capital and small letter:
The sentence has been shorted (Lines 592-594): Rubius Therapeutics, Inc have recently received FDA approval for a Phase Ib clinical trial of genetically engineered erythrocytes expressing phenylalanine ammonia lyase (RTX-134).
3. It is noted that the English grammar, spelling, and sentence structure of the whole manuscript needs careful editing, so that the content of the paper are readable.
3.1 The sentence in line 120 is missing a punctuation.
Full stop has been added (Line 128).
3.2 In line 361, “has been a long interest to Baysal et al” is expressed in wrong way .
The sentence has been amended to (Lines 400-401):The utility of the erythrocyte as a carrier of urease (EC 3.5.1.5) for reducing blood concentrations of urea in patients with chronic renal failure has been a long-term interest of Baysal et al. [44].
3.3 In line 366, it is better to use the phrase “encapsulated in human erythrocytes”.
The sentence has been amended to (Lines 405-407): Urease and alanine dehydrogenase and their pegylated counterparts PEG-urease/PEG-alanine dehydrogenase were successfully encapsulated in human erythrocytes [45,46].
4. The formation of references should be revised according to the rule of the editorial office.
These have been formatted.
5. In “Table 1. Summary of in vitro, preclinical and clinical applications of erythrocyte-mediated enzyme therapy”, “in vitro” perhaps should be the part of preclilnical
Table 1: Preclinical has been changed to in vivo.
Reviewer 2 Report
This is a well-written review paper. Here, Dr. Bax described various studies that utilized the red blood cells as carriers of therapeutic enzymes. The paper is well-organized, and the disease indications discussed are comprehensive. This paper would serve as a good guide for researchers interested in erythrocyte-related therapies. Comments are given below:
- For most part, the paper focused on the benefits of utilizing erythrocyte as enzyme carriers. However, a section on challenges and limitation of erythrocyte-based approaches appear important. For example, clinical translation of RBC-based approaches is often cumbersome (e.g. the need to harvest the RBCs, and formulate the enzyme preparation ex-vivo can be challenging to execute). Likewise, the quality control metrics are hard to standardize on a patient by patient basis. This and many other drawbacks can be discussed, and a perspective can be provided to overcome these barriers.
- It would help to differentiate the approaches that attempt to load enzyme inside RBCs vs the one that aims to attach the enzymes on the erythrocyte membranes. The author did incorporate those studies in the manuscript, but it can be clarified further.
- Erythrocyte based carrier systems have seen several innovations in the recent year. Especially, the use of nanoparticles is beginning to change the landscape for bioscavenging and thrombolytic therapy. Some related references should be incorporated in the paper.
Author Response
The author thanks the reviewer for their constructive feedback. Please find below the author's responses to the queries raised:
1. For most part, the paper focused on the benefits of utilizing erythrocyte as enzyme carriers. However, a section on challenges and limitation of erythrocyte-based approaches appear important. For example, clinical translation of RBC-based approaches is often cumbersome (e.g. the need to harvest the RBCs, and formulate the enzyme preparation ex-vivo can be challenging to execute). Likewise, the quality control metrics are hard to standardize on a patient by patient basis. This and many other drawbacks can be discussed, and a perspective can be provided to overcome these barriers.
The following section has been added to the review:
4. Challenges and limitations of erythrocyte-based enzyme therapy
The limitation of the erythrocyte enzyme carrier is that its metabolic capacity is restricted to the vascular compartment and monocyte-macrophage system. For the former, a further restriction for the encapsulated enzyme, would be the inability of the enzyme’s substrate to permeate the erythrocyte membrane. Thus, the spectrum of clinical applications to which the erythrocyte enzyme carrier can be applied to is fairly narrow. As with most cell-based therapies, the translation of the erythrocyte carrier into the clinical setting faces many complex process-related and regulatory challenges. The enzyme to be encapsulated or conjugated to the erythrocyte membrane will require manufacture in compliance with current Good Manufacturing Practice (cGMP). Although advancements in the field of recombinant DNA biotechnologies have facilitated the manufacture of therapeutic enzymes, many challenging activities exist, including full characterization of cell banks, process- and product-related impurities and enzyme; validation of analytical procedures; formulation design; and stability studies. The transition of the manufacture of the erythrocyte carrier from a laboratory-based scale, to a scale that is clinically relevant is a critical step in the development pathway. Manufacture can be via a decentralised automated process, such as the Red Cell Loader as employed by Erydel or via a centralised process as used by Erytech Pharma[108,109]. Either way, the process will need to be performed in a closed circuit system, using single use, sterile disposables. A key regulatory requirement is the manufacture of a reproducible product. Standardizing the loading procedure (and thus dose) when autologous erythrocytes are employed can be difficult to achieve, and may necessitate the consideration of individualised metrics such as patient haematocrit or red blood cell count in the manufacturing process. To minimise substantial differences in the quality of the end product and to produce a consistent encapsulation rate, Erytech Pharma who employ homologous donated erythrocytes, adjust the process operating conditions such as erythrocyte flow rate and osmolality of buffers according to the osmotic fragility of the initial donor erythrocyte pellet[109]. Prior to clinical batch release by a pharmacist, the product will need to conform to product specifications. For erythrocyte carriers manufactured from homologous blood, the 72-hour shelf-life specified by Erytech provides sufficient time to conduct the necessary tests[109]. However, erythrocyte carriers manufactured from blood collected from autologous blood require a much shorter shelf-life (30 minutes) thus providing a major hurdle to testing. The MHRA approved Phase II trial of EE-TP was able to navigate this issue by proposing a retrospective testing of two of the release parameters. The impact of manufacturing process and chemical modification (where enzymes are conjugated to the erythrocyte membrane) on the function, viability and biocompatibility of the erythrocyte will also need to be established to ensure safety of the medicinal product.
2. It would help to differentiate the approaches that attempt to load enzyme inside RBCs vs the one that aims to attach the enzymes on the erythrocyte membranes. The author did incorporate those studies in the manuscript, but it can be clarified further.
To clarify this the following sentences have been added to the review:
- Lines 157-158: Investigations into this therapeutic application have focused on encapsulating the relevant enzyme within the erythrocyte.
- Lines 289-290: With the exception of phenylketonuria, these therapeutic applications have focused on the use of encapsulated enzyme.
- Lines 603-604: The implementation of the erythrocyte carrier in anti-tumour therapy has focused on the use of encapsulated enzymes and represents…….
3. Erythrocyte based carrier systems have seen several innovations in the recent year. Especially, the use of nanoparticles is beginning to change the landscape for bioscavenging and thrombolytic therapy. Some related references should be incorporated in the paper.
To address this the following paragraph has been added:
More recently, the erythrocyte carrier has been investigated as a theranostic nanoplatform system. This system consists of vesicles derived from erythrocytes that encapsulate the NIR fluorophore, indocyanine green, and of relevance to this review, the conjugation of tPA to the vesicle surface. These constructs are referred to as NIR erythrocyte-derived transducers (NETS). In vitro studies employing a clot model demonstrated the dual functionality of NETS in NIR imaging and clot lysis[107].